# Acute Lymphoblastic Leukemia and Invasive Mold Infections: A Challenging Field

**DOI:** 10.3390/jof8111127

**Published:** 2022-10-26

**Authors:** Christos Stafylidis, Panagiotis Diamantopoulos, Eleni Athanasoula, Elena Solomou, Amalia Anastasopoulou

**Affiliations:** 1First Department of Internal Medicine, Laikon General Hospital, National & Kapodistrian University of Athens, 11527 Athens, Greece; 2Department of Internal Medicine, University of Patras Medical School, 26500 Rion, Greece

**Keywords:** fungal infections, aspergillosis, acute lymphoblastic leukemia, antifungal prophylaxis

## Abstract

Acute lymphoblastic leukemia (ALL) patients comprise a highly immunocompromised group due to factors associated either with the treatment or the disease itself. Invasive mold infections (IMIs) are considered to be responsible for higher morbidity and mortality rates in patients with hematologic malignancies, including ALL. Defining the exact incidence of IMIs in ALL patients has been rather complicated. The available literature data report a highly variable incidence of IMIs, ranging from 2.2% to 15.4%. Although predisposing factors for IMIs in the setting of ALL are ill-defined, retrospective studies have indicated that a longer duration of neutropenia, treatment with high-dose corticosteroids, and a lack of antimold prophylaxis are associated with an increased risk of IMIs. Additionally, the influence of novel ALL treatments on the susceptibility to fungal infections remains obscure; however, initial data suggest that these treatments may induce prolonged neutropenia and thus an increased risk of IMIs. Administering primary antimold prophylaxis in these patients has been challenging since incorporating azole antifungal agents is troublesome, considering the drug-to-drug interactions (DDIs) and increased toxicity that may occur when these agents are coadministered with vincristine, a fundamental component of ALL chemotherapy regimens. Isavuconazole, along with several novel antifungal agents such as rezafungin, olorofim, and manogepix, may be appealing as primary antimold prophylaxis, given their broad-spectrum activity and less severe DDI potential. However, their use in ALL patients needs to be investigated through more clinical trials. In summary, this review outlines the epidemiology of IMI and the use of antifungal prophylaxis in ALL patients.

## 1. Introduction

Invasive fungal infections (IFIs), particularly invasive mold infections, (IMIs) remain one of the leading contributing factors to increased morbidity and mortality among patients with hematologic malignancies [1,2]. Previous reports suggest that the occurrence of IFIs is more common in hematopoietic stem cell transplant (HSCT) recipients and in patients with acute myeloid leukemia (AML), with the latter group being at the highest risk, given the prolonged neutropenia and myeloid cell dysfunction that characterizes AML [1,2,3,4]. While the prevalence and risk factors for IFIs in AML patients are well-defined, data are scarce regarding the epidemiology of IFIs in acute lymphoblastic leukemia (ALL) and derive mostly from studies in pediatric patients [5,6,7]. Moreover, the true burden of IFIs in this population may be hard to estimate, taking into consideration the diversity of ALL treatment protocols and their regional variability in conjunction with the constantly changing epidemiology of IFIs and the lack of reliable and universally approved fungal diagnostic tools. Furthermore, it remains unclarified whether emerging ALL-targeted therapies render these patients susceptible to fungal infections. Although primary antifungal prophylaxis (PAP) against molds has been incorporated in most of AML treatment regimens, no explicit and globally applicable guidelines have been established for ALL [8].

In addition, azole antifungals may interact with vinca alkaloids, a fundamental component of ALL chemotherapy regimens, and increase the risk of induced neurotoxicity, thus hindering their use as prophylactic agents. Hence, an unmet need for suitable antifungal prophylaxis has arisen in these patients. In this report, we review the incidence and predisposing factors for IMIs in adult patients with ALL as well as the currently available strategies for antifungal prophylaxis, while we also investigate whether contemporary ALL-targeted treatments are involved in a higher risk of fungal infection.

## 2. Incidence of IMIs in ALL

ALL represents 15% of leukemias in adults, with annual incidence rates in Europe ranging from 0.9 to 1.3 per 100,000 [9]. Although ALL is less common in adults than in children, it still poses a major threat since it is associated with lower curative rates and inferior survival outcomes, with relative survival rates at 5 years after diagnosis reaching 24% [9]. Adult ALL patients ordinarily undergo highly intensified chemotherapy regimens, consisting of different phases of induction, reinduction, and consolidation, that may last up to three years [9]. Besides the myelosuppression induced by conventional chemotherapy agents, some of these regimens contain high doses of corticosteroids, rendering these patients prone to IMIs. Additionally, during the last decade, increasing awareness of the pathophysiology and the genomic landscape of ALL and the recognition of new molecular targets ushered in an era of promising novel targeted agents used in the treatment of ALL [10]. However, their risk of infectious complications and their interactions with azole antifungals remain unclear [11,12].

Epidemiological data regarding the incidence rate of IMIs, and IFIs in general, among ALL patients remain obscure [7]. Most studies have been conducted in pediatric ALL patients, reporting vastly fluctuating incidences of IFIs, ranging from 0.01% to 22%. However, a direct comparison with adult ALL and drawing conclusions should be avoided, given the discrepancies in the genetic lesions of ALL, age, and pre-existing comorbidities between adult and child patient groups [7].

As seen in Table 1, most of the studies in adult ALL patients report quite variable incidence rates of proven/probable IFIs, within a range of 4.3% to 18.3% [1,7,13,14,15,16,17,18,19,20,21,22,23,24,25]. Accordingly, the frequency of IMIs in this group is highly fluctuating within a range of 2.2% to 15.4% [1,14,15,16,17,18,20,21,22,23,25]. The great variability in the incidence reports among these studies reflects the heterogeneity of patient populations with regards to the presence of risk factors for IMIs as well as differences in study designs/applied diagnostic criteria.

## 3. The Impact of Novel Diagnostic Tools/Criteria and Antimold Prophylaxis on IMI Incidence in Adult ALL Patients

A definitive IMI diagnosis is demanding and requires the identification of the pathogen in the affected tissue. Given that the risk of complications related to biopsy is high in hematological patients, diagnosis is frequently based on a constellation of host factors and clinical and mycological criteria. The 2008 criteria defined by the European Organization of Research and Treatment of Cancer and the Mycoses Study Group (EORTC/MSG) classify IMIs as proven, probable, and possible and were recently updated [26]. Although these criteria are now utilized by most medical faculties, they were not implemented in earlier studies [26]. Recent studies have assessed the impact of novel diagnostic tools and criteria on the estimation of the incidence of IMIs. In a Brazilian study conducted in the period before and after 2008, when testing for galactomannan (GM) in their center became available, a higher risk of developing a mold infection was reported after 2008 [15]. Moreover, a recent retrospective trial evaluated the impact of novel EORTC/MSG criteria on patient classification and reported 11.1% more probable invasive pulmonary aspergillosis (IPA) diagnoses, mostly due to positive polymerase chain reaction (PCR) results [27]. By contrast, in another study, 27.3% of probable cases were reclassified as possible due to novel GM cut-off levels [28].

Moreover, the influence of antifungal prophylaxis on the incidence of IMIs remains undefined. While mold infections have long been considered more frequent in AML patients [1,2,3,4], the inclusion of mold-active azole prophylaxis during the induction and reinduction phases resulted in reduced infection rates [29]. In a retrospective study, the overall frequency of IPA was similar in AML and in ALL patients (16.2% and 15.4%, respectively). The same study also demonstrated that most of the IPAs in ALL patients were proven/probable, while almost half of them were possible in AML patients, with the relative frequency of proven/probable IPA being higher in ALL than AML patients (83% vs. 54.5%) [20]. Moreover, one retrospective study demonstrated that the incidence of IFIs in AML patients with antimold prophylaxis was not higher than that in ALL without antimold prophylaxis, reflecting the need to re-evaluate the risk in ALL patients and to find a suitable antifungal agent [25]. In the same study, an IMI incidence rate of 4.7% was reported in ALL patients without antimold prophylaxis versus 0% in ALL patients receiving antimold prophylaxis [25]. Additionally, an Australian study exhibited a tremendously higher incidence of proven/probable IFIs in ALL patients not receiving PAP than in those receiving it (21.4% vs. 2.6%, *p* = 0.02) [14]. On the whole, these studies, given their retrospective nature and the small and highly heterogeneous population samples, reflect the need to design further prospective trials in order to assess an accurate rate of IMIs in ALL and the impact of antimold prophylaxis as well as to recognize risk factors that are associated with a great risk of developing a fungal infection.

## 4. Risk Factors for IMIs in Adult ALL Patients

Despite the lack of large prospective clinical trials, some studies have described factors that put ALL patients at a higher risk of an IFI. O’Reilly et al. showed in a retrospective study that a shorter course of dexamethasone and a less intensified induction regimen were associated with fewer IFIs [7]. Another retrospective study revealed that the duration of neutropenia and a lack of antimold prophylaxis were independent risk factors for developing an IFI [25]. In a large population-based study of patients with hematologic malignancies, which mostly consisted of ALL patients, neutropenia, increasing age, acute renal failure, hemodialysis, viral and *Clostridium difficile* infections, admission to an intensive care unit (ICU), and residency in a rural area were all correlated with a significantly higher risk of developing an IFI [22]. Risk assessment for mold-related infections has not been widely performed among ALL patients, but a variety of studies have identified risk factors for IMIs in patients with hematologic malignancies and immunocompromised patients in general, which could also be applied at a theoretical level to ALL patients. It has long been established that neutropenia lasting for more than three weeks and treatment with high-dose corticosteroids for over a week render immunocompromised patients prone to IMIs [2,7,22,25]. Most of the crucial factors, which are either host-related, leukemia-related, or associated with fungal exposure, are summarized in Table 2 [2,7,14,22,25,30,31,32,33,34,35,36,37,38,39,40,41,42].

Although some risk models for IMIs have been developed during the past years, there is no current universally applied model for ALL patients. The D-index was proposed by a team of investigators to measure the risk of IMIs in neutropenic AML patients and seemed to have a high negative predictive value (97–99%) [43]. However, data regarding its use in ALL patients are insufficient. More recently, Stanzani and Lewis developed a risk model for IMIs in patients with hematologic malignancies, the revised BOSCORE model, which is quite easy to use, and the results are promising [44]. However, the benefit of its application in daily clinical practice remains to be validated. Nonetheless, further efforts should be made in developing an ideal risk model that could result in the early recognition of those who are at risk and an individualized approach.

## 5. Risk of IMIs in the COVID-19 Era

Early in the course of the pandemic, COVID-19-associated pulmonary aspergillosis (CAPA) was primarily reported in critically ill and/or mechanically ventilated patients, resulting in substantially high 30-day mortality rates of up to 44% [45]. COVID-19-associated mucormycosis (CAM) was initially reported in Indian patients with uncontrolled diabetes, in whom it most commonly presented as rhino-orbital cerebral mucormycosis, whereas pulmonary disease occurred almost exclusively in ICU patients [46]. The exact immunopathogenesis of CAPA and CAM is unknown, but is likely multifactorial, including viral factors, fungal factors, and the host immune response [47]. Moreover, the exact incidence of these entities is unknown and may be overestimated due to the lack, until recently, of standardized diagnostic criteria [48]. In ALL patients, who have established risk factors for the development of IFIs, it is unknown whether COVID-19 per se may represent an additional risk factor. In a retrospective single-center study of 46 patients with COVID-19 infection and acute leukemia (including 10 ALL patients), probable CAPA was diagnosed in 22% [49]. Among pediatric patients with cancer and COVID-19, 10.5% were diagnosed with CAPA and 70% of them had hematologic malignancies. Importantly, the majority of patients (65%) were taking antifungal prophylaxis with caspofungin or anidulafungin upon CAPA diagnosis [50]. Although further research is needed, clinicians should be aware of the potential risk of COVID-19-associated fungal infections.

## 6. Risk of IMIs with Novel ALL Treatments

During the past years, novel targeted treatments have dramatically altered the management of relapsed/refractory ALL, while their use is being evaluated in earlier stages of the disease [10]. Although these agents are considered to be less myelosuppressive than conventional chemotherapy, their infectious risk for IMIs, and IFIs in general, remains undefined. Evaluating the accurate risk of novel agents for IMIs is complicated since patients receiving these treatments are immunocompromised by default, and preceding or concurrent immunosuppressive therapies may further hinder an exact estimation.

Inotuzumab ozogamycin (InO) is a CD22-directed antibody–drug conjugate that is indicated for the treatment of relapsed/refractory CD22-positive B-cell precursor ALL [10]. In the INO-VATE study, which compared ALL patients receiving InO monotherapy versus standard of care (SoC) intensive chemotherapy, there were similar rates of treatment-related severe neutropenia in both arms (36% vs. 37.8%), while less febrile neutropenic events were observed in the InO arm (26.8% vs. 53.8%), with cases of fungal pneumonia being diagnosed only in the SoC arm [51]. Another phase 2 study showed that the use of InO in combination with low-intensity chemotherapy in older ALL patients, though safe, was associated with a high rate of severe infections (92.3%) and a prolonged duration of neutropenia (median recovery duration of 16 days) [52]. However, the incidence of IMIs was not reported in this study. Although data are lacking regarding the risk of InO for IMIs, prophylaxis against molds should be considered in these patients, given the reported prolonged neutropenia periods.

Blinatumomab is a bispecific T-cell engager (BiTE) antibody that redirects CD3-positive cytotoxic T cells to lyse CD19-positive B cells and is also used in the treatment of relapsed/refractory B-precursor adult ALL [10]. The TOWER trial, which compared blinatumomab and SoC chemotherapy in adult patients with refractory/relapsed ALL, demonstrated a reduced incidence of neutropenia in the blinatumomab group (37.8% vs. 57.8% in the SoC arm) along with a lower incidence of overall infection (34.1% vs. 52.3%) [53]. In the aforementioned trial, the reported incidence of IFIs was 3.7% in the blinatumomab arm, which was lower than that in the SoC arm, and the majority of these infections were attributed to molds [53]. In another trial, IFIs were reported in 11 out of 189 (5.8%) patients while receiving blinatumomab treatment [54]. Although blinatumomab is considered less myelosuppressive than SoC chemotherapy, it could still make ALL patients prone to IMIs since CD19 is a major component of the host’s defense against fungi by intermingling with signaling thresholds, affecting the B-cell-dependent activation of T cells, and inducing profound hypogammaglobulinemia [55].

CD19-targeted chimeric antigen receptor-modified T (CAR-T)-cell immunotherapy is another novel treatment for relapsed/refractory ALL. CAR-T-cell recipients are vulnerable to infections due to CAR-T-cell-associated toxicities, such as neutropenia, which can have a long duration of up to 60 days, as was estimated in a previous study, and is probably attributed to disruptions in chemokines involved in neutrophil trafficking [56]. Hence, these patients are at a high risk of infection, not only during the first days postinfusion but also for a quite prolonged period of up to 90 days [56]. Moreover, preceding neutropenia and lymphocyte depletion due to conditioning regimens or prior HSCT may further increase infectious risk. CAR-T-cell treatment is also frequently complicated with cytokine release syndrome (CRS) and CAR-T-cell-related encephalopathy syndrome (CRES), which usually require the administration of tocilizumab, an interleukin-6 (IL-6) receptor inhibitor, and dexamethasone, thus importantly elevating the infectious risk. Data regarding the IMI incidence after CAR-T-cell therapy in ALL patients are still limited. However, some small studies demonstrated an IMI incidence rate of 1–7% [57,58,59]. The impact of antimold prophylaxis is yet to be examined. However, it should be strongly considered in this setting. Though there are no consensus guidelines for antimold prophylaxis in ALL patients after CAR-T-cell treatment, it is recommended by the 2018 American Society of Clinical Oncology and Infectious Diseases Society of America (ASCO/IDSA) guidelines to administer antimold prophylaxis in adults with cancer if the population level risk of aspergillosis is ≥6% [60]; this threshold may be applied to ALL CAR-T-cell recipients as well since a few studies reported an IMI incidence of 7%, as previously mentioned. On the other hand, the use of antimold prophylaxis may be hampered by its high cost, its induced toxicities, and the emergence of breakthrough IMIs caused by resistant molds [59]. Hence, until rigorous, prospective, multicenter trials are conducted to assess the IMI rates in ALL CAR-T-cell recipients, the administration of antimold prophylaxis should be considered in individual patients suffering from neutropenia for more than three weeks pre- or post-CAR-T-cell infusion, in patients complicated by CRS/CRES, in those receiving corticosteroids for ≥7 days, and in institutions where IMI rates are ≥6%. 

## 7. Antimold Prophylaxis in ALL

Despite the integration of azole antifungals into treatment regimens in the majority of patients with hematologic malignancies, their use in ALL patients remains limited, mostly due to drug-to-drug interactions (DDIs) [61]. Vincristine is an essential component of most of the chemotherapy regimens used in ALL, and its metabolism is mainly performed by the cytochromes CYP3A4 and CYP3A5. Azole antifungals, each one possessing differing inhibitory capabilities, lead to the inhibition of CYP3A, thus enhancing vincristine-induced toxicity [61]. However, pharmacokinetic data are lacking to further support this hypothesis. Moreover, azole antifungals, when used long-term, may have neurotoxic properties themselves [61]. Though no data regarding the concomitant use of azole antifungals and vincristine in adults with ALL are available, investigators have described an augmentation of vincristine-induced toxicity in pediatric ALL patients with simultaneous azole intake [62]. Accordingly, in another retrospective study that compared vincristine’s toxicity between azole- and non-azole-treated children with ALL, it was reported that constipation and peripheral neurotoxicity were significantly more frequent in patients receiving vincristine in combination with azole treatment, while vincristine-induced central nervous system (CNS) toxicity occurred only in this group [63]. Opposingly, another study in pediatric ALL patients demonstrated no difference in the incidence of grade 2 or greater neuropathy with the concurrent use of antifungal therapy, whereas the incidence was found to be higher only with increasing doses of vincristine [64]. A recent randomized controlled trial that evaluated vincristine-induced peripheral neurotoxicity in pediatric ALL patients by comparing push injections of vincristine with one-hour infusions showed that toxicity was less severe in patients who received vincristine as a one-hour infusion compared to those who received it as a push injection [65]. Hence, although clinical trials have not been conducted in the setting of adult ALL, physicians should consider the administration of vincristine as a one-hour infusion whenever antimold prophylaxis is imperative.

Besides vincristine, the coadministration of azoles with some other chemotherapy agents may be responsible for DDI appearance. Cyclophosphamide, which is commonly used in ALL treatment protocols, is primarily metabolized through CYP3A4 and CYP2C9 [66]. An open-labeled randomized trial in allogeneic HSCT patients demonstrated that fluconazole, through the inhibition of CYP2C9, and itraconazole, through the inhibition of CYP3A4, induced higher plasma concentrations of cyclophosphamide, and a higher rate of abnormal values in serum bilirubin and creatinine was observed in patients simultaneously receiving these agents [66]. Due to azoles’ ability to inhibit liver cytochromes and p-glycoprotein (p-Gp), they could interact at a theoretical level with more drugs as their metabolism is carried by these pathways and result in enhanced toxicity. For example, anthracyclines and tyrosine kinase inhibitors, which are also used in Philadelphia-positive ALL treatment, are metabolized by CYP3A4 [67]. Moreover, since doxorubicin is effectively effluxed from the cell via p-Gp, the administration of certain azoles could possibly lead to high intracellular levels of these drugs and toxicity [67]. However, evidence is needed to further support these assumptions. Finally, azole antifungals could possibly interact with novel targeted ALL treatments. For instance, InO, which can generate prolonged neutropenia, making the use of PAP quite necessary, causes the prolongation of the QTc interval [12]. Therefore, concomitant use with azoles, which also prolong QTc, could prompt the appearance of an arrhythmia. Conjointly, all of these DDIs render the employment of azoles as PAP in ALL patients troublesome.

Isavuconazole, a novel antifungal azole that displays excellent activity against yeasts and molds, may be an appealing candidate for antimold prophylaxis in ALL patients since it appears to have fewer serious adverse events and is a moderate CYP3A inhibitor with less DDIs in comparison with other azoles [68]. Importantly, isavuconazole shortens the QTc interval, and this effect has not been associated with adverse cardiac events [69]. Although, isavuconazole is indicated for the treatment of invasive aspergillosis and mucormycosis, its efficacy as an antimold prophylaxis has not been extensively examined, let alone in adult ALL patients. Data from previous studies in patients with hematologic malignancies and HSCT or solid organ transplant recipients are equivocal but encouraging [70,71,72,73,74]. A study that evaluated the efficacy and safety of isavuconazole compared with voriconazole as PAP in HSCT recipients demonstrated that isavuconazole was equally effective and better tolerated than voriconazole [70]. Correspondingly, a similar study in patients following lung transplantation showed comparable efficacy and a better safety profile of isavuconazole compared to voriconazole [71]. An open-label prospective phase 2 study in patients with AML or myelodysplastic syndrome (MDS) demonstrated that isavuconazole was safe and effective as an alternative for PAP in patients receiving remission-induction chemotherapy [72]. On the contrary, a large retrospective study of patients with hematologic malignancies and HSCT recipients failed to show a benefit from isavuconazole prophylaxis, whereas it also reported a higher rate of breakthrough IFIs, with a notable rate of IPA [73]. Recently, a systematic review highlighted isavuconazole’s effectiveness and safety as a PAP [74]. Although these data could be extrapolated in the context of ALL, more studies need to be performed to elucidate isavuconazole’s efficacy as a primary antimold prophylaxis in ALL patients. 

Echinocandins could be an attractive alternative choice of PAP in ALL patients, given their favorable safety profile and their low potential for DDIs since they do not exert inhibitory effects on the CYP3A system [75]. Although echinocandins are fungistatic against *Aspergillus* spp., most species are susceptible in vitro [76], whereas they do not show activity against *Fusarium* spp. and Mucorales. Several studies have examined the safety and efficacy of their use as prophylaxis, mostly in AML and HSCT patients, with various results [75,77]. Amphotericin B and its lipid formulations have the widest spectrum and are active against both yeasts and molds, including agents of mucormycosis. Moreover, besides their nephrotoxic properties, they lack DDIs through the CYP3A system [78]. However, the AmBiGuard study, a randomized double-blind multicenter clinical trial that evaluated the efficacy of liposomal amphotericin B (L-AMB) in adult ALL patients undergoing remission-induction chemotherapy, showed no statistically significant difference in the incidence of IFIs between the two arms (7.9% in the arm receiving L-AMB vs. 11.7% in the placebo arm, *p* = 0.24) [19]. Nonetheless, more randomized clinical trials are required to assess the efficacy of echinocandins and amphotericin B as PAP in the setting of ALL.

## 8. Novel Antifungal Agents

Over the last few years, the armamentarium of antifungals has been expanding, and some novel drugs may be quite promising. Opelconazole, a novel inhaled triazole, which demonstrates efficacy primarily in the lungs, has a low potential for DDIs and systemic adverse effects and exerts a wide-spectrum antifungal activity against yeasts and molds, including Rhizopus spp. [79]. Moreover, opelconazole persists in local immune and epithelial cells. Hence, this ability renders it an appealing agent for prophylactic use [79]. Rezafungin, a second-generation echinocandin with enhanced pharmacokinetics and pharmacodynamics and a favorable safety profile, exhibits potent activity against *Candida* spp., *Pneumocystis jirovecii*, and *Aspergillus* spp. [79]. Lately, rezafungin’s efficacy as a prophylaxis was evaluated in vivo in mouse models of invasive aspergillosis, invasive candidiasis, and Pneumocystis pneumonia, and the results are quite encouraging, thus establishing the basis for further investigation of rezafungin as a PAP in clinical trials [80].

Olorofim is a first-in-class agent that inhibits fungal dihydroorotate dehydrogenase, a key enzyme of pyrimidine synthesis [79,81]. It can be administered orally and is metabolized by cytochrome P450 isoenzymes, while it also has a weak inhibitory potency for CYP3A4, which could prevent DDIs [79,81]. Olorofim, despite not being a broad-spectrum antifungal since it is not active against yeasts such as *Candida* spp. or the Mucorales group, exerts some significant activity against *Aspergillus* spp. and dimorphic and dematiaceous molds, including multiresistant molds [79,81]. Hence, in the future, olorofim could prove to be a valuable weapon in the prophylaxis and treatment of IMIs. Finally, manogepix is an orally available novel agent that inhibits the fungal enzyme Gwt1, thus leading to the disruption of glycosylphosphatidylinositol-anchored protein maturation [79,82]. This action seems to be fungal-pathogen-specific, thus minimizing its toxicity potential. Furthermore, manogepix is accompanied by a favorable safety profile and no serious DDI potential, while it has wide tissue distribution with no food effect on its absorption [79,82]. Regarding its activity, manogepix is quite an appealing agent since it has one of the broadest-spectrum activities, including *Candida* spp., *Aspergillus* spp. *Fusarium* spp., *Scedosporium* spp., *Lomentospora prolificans*, *Rhizopus* spp., and other rare molds, and shows potent activity against resistant molds [79,82]. Currently, several ongoing clinical trials are investigating manogepix’s safety and efficacy in the treatment of IFIs in various clinical settings, such as AML, and hopefully more trials will be conducted to examine its role as a PAP. All things considered, these novel antifungal agents could be employed as PAP, while the crucial need to develop an ideal antifungal prophylaxis agent for ALL patients should sow the seeds for further research.

## 9. Conclusions

The incidence of IMIs and the role of antimold prophylaxis in ALL has not been thoroughly evaluated yet since data are sparse and there are no large randomized controlled trials. Moreover, more light should be shed on previously discussed factors that put ALL patients at a greater risk of IMIs in order to recognize those that would benefit from antimold prophylaxis. Despite the fact that no consensus guidelines exist, physicians should particularly consider the administration of antimold prophylaxis in ALL patients that have the aforementioned risk factors, especially those who are neutropenic for more than three weeks and those receiving high-dose corticosteroids for over a week. Although the choice of antifungal agent for antimold prophylaxis in ALL is rather complex, given the risk of the aforementioned DDIs, especially when using azoles, several new agents with extended-spectrum activity and fewer DDIs are quite promising, and their use as primary antimold prophylaxis in this subset of patients should be investigated in more prospective clinical trials.

## Figures and Tables

**Table 1 jof-08-01127-t001:** Studies in patients with acute lymphoblastic leukemia reporting the incidence of invasive fungal infections and invasive mold infections.

	Study Type	Study Period (years)	Criteria for IFI Diagnosis	Patients (N)	Median Age (Range)	Type of Treatment	Corticosteroid Use	Antifungal Prophylaxis	Overall IFI Incidence, %	IMI Incidence, %	Comments
Pagano et al. 2004 [1]	Retrospective, multicenter	4 (1999–2003)	2002 EORTC/MSG	1173	NR	NR	NR	Was administered in some centers but it is not reported which patients among the different subgroups received it	6.5 proven/probable	4.3 proven/probable	44 cases of *Aspergillus* spp., 4 cases of Zygomycetes, 1 case of *Fusarium* spp., and 2 undefined IMIs
Henden et al. 2013 [13]	Retrospective, single institution	5 (2005–2010)	2008 revised EORTC/MSG	32	28 (15–72)	Hyper-CVAD	Yes	All patients received fluconazole	28 9.7 proven 3.2 probable 19.4 possible	NR	- One patient suffered from two separate episodes of IFI - Among 3 proven episodes of IFI, 2 were attributed to *Scedosporium prolificans.*
Doan et al. 2016 [14]	Retrospective, multicenter	5 (2008–2013)	2008 revised EORTC/MSG	98	43 (29–57)	Hyper-CVAD, BFM95, LALA94, ALL6, ANZCHOG Study 8, and others	Yes	85% in total 65% L-AMB 18% posaconazole 8% fluconazole 5% caspofungin 4% voriconazole	5.1 proven/probable 6.1 possible	4.1 proven/probable	Statistically significant lower incidence of proven/probable IFIs in patients receiving PAP versus those not receiving PAP (2.6% vs. 21.4%, *p*: 0.02)
Nicolato et al. 2016 [15]	Retrospective, single institution	26 (1987–2013)	2008 revised EORTC/MSG	153	24 (12–75)	BFM protocols, Hyper-CVAD, HiDAC for relapsed ALL, Allogeneic HCT, Autologous HCT	Yes	Antifungal prophylaxis, including mold-active azoles, was given in 27.4% of episodes of febrile neutropenia	18.3 proven/probable	NR	- Prevalence of proven/probable IFIs 8.8% - Prevalence of IMIs 61.3% (19/31 IFIs) - 32.2% of the IFI episodes in patients receiving prophylaxis with an azole
Mariette et al. 2017 [16]	Retrospective, multicenter	6 (2006–2012)	2008 revised EORTC/MSG	969	47	GRAALL-2005, GRAALL-R-2005, GRAAPH-2005	Yes	No standard use	7.8	3.3 IA 0.2 *Fusarium* sp. 0.1 Zygomecetes 0.1 *Scedosporium* sp.	15.4% of patients with IA, 21.2% of patients with IC, and 50% of patients with other IFI were taking antifungal prophylaxis.
Keng et al. 2017 [17]	Retrospective, single institution	14 (1999–2013)	2008 revised EORTC/MSG	209	NR	Hyper-CVAD, HKSG ALL 97, FLAG, HiDAC	Yes	Itraconazole L-AMB posaconazole caspofungin fluconazole	11.5 proven/probable	4.3 proven IA 0.9 proven *Fusarium* sp. 0.9 proven *Rhizopus* sp.	All patients with an IFI episode were receiving antifungal prophylaxis.
Koehler et al. 2017 [18]	Prospective, multicenter	2 (2011–2013)	2008 revised EORTC/MSG	627	NR	NR	NR	54.2% received antifungal prophylaxis: Posaconazole Fluconazole L-AMB		3.8 proven/probable IA	Incidence of IA was higher in AML patients (6.4%).
Cornely et al. 2017 [19]	Randomized, double-blind, multicenter phase 3 trial	2 (2011–2013)	2008 revised EORTC/MSG	339 (included in the efficacy analysis)	45 (32–57) in the L-AMB group and 47 (28–60) in the placebo group	NR	NR	67.3% 5 mg/kg L-AMB twice weekly 32.7% placebo	7.9 proven/probable and 4.8 possible in the L-AMB group 11.7 proven/probable and 5.4 possible in the placebo group	NR	L-AMB showed no benefit in preventing IFIs in ALL patients receiving remission-induction chemotherapy.
Cattaneo et al. 2017 [20]	Retrospective, single institution	5 (2011–2016)	2008 revised EORTC/MSG	39	NR	NILG 10/07 GIMEMA protocol	NR	All ALL patients received fluconazole prophylaxis		15.4 IPA	- The incidence of IMIs includes proven/probable and possible cases. - The overall frequency of IPA was similar in AML (16.2%) and in ALL (15.4%) patients. - Relative frequency of proven/probable IPA was higher in ALL (83%) than in AML (54.5%).
Di Blasi et al. 2018 [21]	Prospective, multicenter	1 (2012–2013)	2008 revised EORTC/MSG	271	46 (19–75)	NILG 10/07, GIMEMA, GMALL	Yes	One of the following antifungal agents was administered as prophylaxis during remission-induction chemotherapy in 37 cases: Fluconazole (19 cases) Itraconazole (1 case) Posaconazole (4 cases) L-AMB (13 cases)	4.3 (17 IFIs/395 treatment cycles)	2.2 proven/probable IA	Coinfections were present in 10 IFI episodes (9 cases with bacteria and 1 with both bacteria and viruses). Combined incidence of IFI was 6.8%.
O’ Reilly et al. 2019 [7]	Retrospective, multicenter	4 (2013–2017)	2008 revised EORTC/MSG	275	5 (1–24)	UKALL2011	Yes	No	8 proven/probable/possible 2.5 proven 1.5 probable 4 possible	0.7 proven/probable	Only 1 proven case of Scedosporium and 1 probable case of Aspergillus niger were reported.
Valentine et al. 2019 [22]	Retrospective, multicenter	11 (2015–2016)	NR	669	NR	NR	NR	NR	11	4.37 IA 0.75 Mucormycosis	
Paige et al. 2019 [23]	Retrospective, single institution	2 (2014–2016)	2008 revised EORTC/MSG	14	NR	Hyper-CVAD and other protocols	Yes	Yes, according to local protocols	14.2 proven/probable	14.2 proven/probable	Both of the two patients received voriconazole prophylaxis.
Grundahl et al. 2020 [24]	Retrospective, single institution	10 (2005–2015)	2008 revised EORTC/MSG	58	49.2 (17–87)	GMALL	Yes	44.8% in total Fluconazole 71.9%	12.1 proven/probable 3.4 proven 8.6 probable 17.2 possible	NR	All patients with proven and 3 out of 5 patients with probable IFIs did not receive antifungal prophylaxis.
Sang-min Oh et al. 2021 [25]	Retrospective, single institution	2 (2017–2019)	2008 revised EORTC/MSG and 2020 EORTC/MSG	51	45 (31–62)	NR	NR	6.9% antimold prophylaxis 3.1% posaconazole, 3.8% other antifungals	7.5 8.1 without prophylaxis 0 with prophylaxis	4.4 IA 0 with antimold prophylaxis 4.7 without antimold prophylaxis	- 159 episodes were reported. - Episode was defined as the period between start of chemotherapy until discharge. - Same incidence results when 2020 EORTC/MSG criteria were applied Incidence of IFI in ALL without prophylaxis was not lower than in AML with antimold prophylaxis.

ALL: acute lymphoblastic leukemia, IFI: invasive fungal infection, IMI: invasive mold infection, EORTC/MSG: European Organization for Research and Treatment of Cancer/National Institute of Allergy and Infectious Diseases Mycoses Study Group, N: number, NR: not reported, hyper-CVAD: hyperfractionated cyclophosphamide, vincristine, doxorubicin, and dexamethasone, BFM: Berlin–Frankfurt–Münster protocol, LALA94: Leucémie Aigüe Lymphoblastique de l’Adulte ’94, ANZCHOG: Australian and New Zealand Children’s Oncology Group, HiDAC: high-dose cytarabine, HCT: hematopoietic cell transplantation, GRAALL: Group for Research on Adult Acute Lymphoblastic Leukemia, GRAALL-R: Group for Research on Adult Acute Lymphoblastic Leukemia-Rituximab, GRAAPH: Group for Research on Adult Acute Lymphoblastic Leukemia Philadelphia-positive, HKSG 97: HongKong Singapore 1997 protocol, FLAG: fludarabine, cytarabine, and filgrastim, NILG 10/07: Northern Italy Leukemia Group 10/07 protocol, GIMEMA: Italian Group for Adult Hematologic Diseases, UKALL 2011: United Kingdom National Randomised Trial for Children and Young Adults with Acute Lymphoblastic Leukaemia and Lymphoma 2011, GMALL: German multicenter acute lymphoblastic leukemia protocol, L-AMB: liposomal amphotericin B, IPA: invasive pulmonary aspergillosis, IA: invasive aspergillosis, PAP: primary antifungal prophylaxis, AML: acute myeloid leukemia.

**Table 2 jof-08-01127-t002:** Risk factors associated with a higher risk of invasive mold infection in hematologic patients.

HOST-RELATED
Genetic factors
Polymorphisms in genes regulating IL-10 production (ACC/ATA and ATA/ATA haplotypes) [30] Polymorphisms in genes regulating the expression of TNFa receptor [31] Polymorphisms in TLR-4 [32] Polymorphisms in plasminogen alleles [33] Mannose-binding lectin deficiency [34] Dectin-1 deficiency [35]
Advanced age [22]
Comorbidities
COPD [36,37] Acute renal failure [22] Hemodialysis [22] Respiratory viral infection [22] Clostridium difficile infection [22]
Admission to an ICU [22]
Smoking [37,38]
**LEUKEMIA-RELATED**
Degree and duration of neutropenia [22,25,37]
Leukemia status (relapse/refractory > first induction > consolidation) [37]
Lower probability of CR [37]
Adverse cytogenetic/gene mutation profiles Immunophenotype WBCs ≥ 30.000/μL
Lymphocytopenia [37]
Treatment-associated
Corticosteroid use (dose and duration) [2,7,37] Intensified induction regimens [7] Use of antimold prophylaxis [14,25] Iron overload [39]
**FUNGAL EXPOSURE**
Prior aspergillosis and/or airway colonization by *Aspergillus* spp. [37]
Dry weather with high temperatures (summer and autumn) [40]
Residency in rural areas [22]
Occupation, e.g., farmers [41]
Building activities in the hospital [37]
Absence of HEPA-filtered rooms [42]
Contamination of water supplies [37]

IL-10: interleukin-10, TNF-a: tumor necrosis factor a, TLR-2: Toll-like receptor 2, TLR-4: Toll-like receptor 4, COPD: chronic obstructive pulmonary disease, ICU: intensive care unit, CR: complete remission, WBC: white blood cell, HEPA: high-efficiency particulate air.

## Data Availability

Not applicable.

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
