# Peer review of "Acute Lymphoblastic Leukemia and Invasive Mold Infections: A Challenging Field"

_jof, 2022, doi:10.3390/jof8111127_

Round 1

Reviewer 1 Report

Stafylidis and colleagues present a comprehensive but still detailed overview of the literature concerning invasive mould infections (IMI) in ALL patients. The review is well-written, well-organized, and very informative.

As only minor criticism, it must be stated that some paragraphs are not focusing on IMI in ALL but are more general, for instance the section about novel antifungal drugs. This is, of course, due to the lack of data / studies specific for ALL patients, ant these sections are still beneficial for readers interested in the new developments in the fields of IMI and antifungal therapy.

Minor comments

Ll.46-47: Please revise this sentence.

Ll.95-98: The statements seem to address the diagnosis of definitive IMI rather than definitive IFI, e.g., candidemia.

L.147 / table 2: The use of bold letters should be explained.

Ll.204-206: What was the incidence in the SoC arm? Higher or lower than for blinatumomab?

Ll.206-207: What was the incidence in the SoC arm? Higher or lower than for blinatumomab?

Author Response

Dear Reviewer,

Thank you for reviewing our manuscript and for giving us an opportunity to revise our work. We have revised our manuscript jof-1979500 entitled “Acute lymphoblastic leukemia and invasive mould infections. A challenging field” in accordance with your comments. You can view all changes in the revised manuscript using the “Track Changes” function.

We would like to thank you for your time spent reviewing our manuscript and for your helpful and constructive comments which have improved it. We have carefully addressed, point-by-point, the concerns raised in the comments by you. Our responses are highlighted in red as follows.

Stafylidis and colleagues present a comprehensive but still detailed overview of the literature concerning invasive mould infections (IMI) in ALL patients. The review is well-written, well-organized, and very informative.

As only minor criticism, it must be stated that some paragraphs are not focusing on IMI in ALL but are more general, for instance the section about novel antifungal drugs. This is, of course, due to the lack of data / studies specific for ALL patients, ant these sections are still beneficial for readers interested in the new developments in the fields of IMI and antifungal therapy.

Author Response: Thank you for your kind words regarding the manuscript. We agree with your comment that some paragraphs are general rather than focusing on IMI in ALL and that is why we highlight in the manuscript several times that there is lack of data specific for ALL patients. Moreover, regarding novel antifungal drugs we specify that these agents could be employed as PAP in the setting of ALL, but further research is needed.

Minor comments

Ll.46-47: Please revise this sentence.

Author reply: Revised as requested. The sentence now reads: “Furthermore, it remains unclarified, whether emerging ALL targeted therapies render these patients susceptible to fungal infections.”

Ll.95-98: The statements seem to address the diagnosis of definitive IMI rather than definitive IFI, e.g., candidemia.

Author reply: Thank you. Revised as requested.

L.147 / table 2: The use of bold letters should be explained.

Author reply: Thank you. We removed the bold letters and we reformatted the table in order to avoid confusion.

Ll.204-206: What was the incidence in the SoC arm? Higher or lower than for blinatumomab?

Author reply: Thank you. The sentence now reads: “a reduced incidence of neutropenia in the blinatumomab group (37.8% vs. 57.8% in the SoC arm)

Ll.206-207: What was the incidence in the SoC arm? Higher or lower than for blinatumomab?

Author reply: The incidence was higher in the Soc arm. The sentence now reads: “In the aforementioned trial, the reported incidence of IFIs was 3.7% in the blinatumomab arm, which was lower than that in the SoC arm, and..”

Reviewer 2 Report

This review paper by Stafylidis et al. perfectly fits the Special Issue "Diagnostic and Therapeutic Challenges of Human Fungal Infections". The authors highlighted the gap in literature regarding invasive mould infections and appropriate prophylaxis in adult ALL patients. Initially, they summarized in Table 1 the reported incidence of IFIs and IMIs, pointing out their variability among different studies. Then went on discussing different risk factors, also dedicating one section to COVID-19-associated mould infections. Finally, they extensively reviewed the challenges of administering antimould prophylaxis in ALL patients, focusing on drug-to-drug interactions, and the promising novel antifungal agents.

This is an interseting review that would be of help to clinicians working with adult ALL patients. However, both tables need to be revised in order to more accurately present the data of the cited papers. Please see the attached file for specific comments on the shortcomings of the manuscript and the tables in particular.

Author Response

Dear Reviewer,

Thank you for reviewing our manuscript and for giving us an opportunity to revise our work. We have revised our manuscript jof-1979500 entitled “Acute lymphoblastic leukemia and invasive mould infections. A challenging field” in accordance with your comments. You can view all changes in the revised manuscript using the “Track Changes” function.

We would like to thank you for your time spent reviewing our manuscript and for your helpful and constructive comments which have improved it. We have carefully addressed, point-by-point, the concerns raised in the comments by you. Our responses are highlighted in red as follows.

This review paper by Stafylidis et al. perfectly fits the Special Issue "Diagnostic and Therapeutic Challenges of Human Fungal Infections". The authors highlighted the gap in literature regarding invasive mould infections and appropriate prophylaxis in adult ALL patients. Initially, they summarized in Table 1 the reported incidence of IFIs and IMIs, pointing out their variability among different studies. Then went on discussing different risk factors, also dedicating one section to COVID-19-associated mould infections. Finally, they extensively reviewed the challenges of administering antimould prophylaxis in ALL patients, focusing on drug-to-drug interactions, and the promising novel antifungal agents.

This is an interseting review that would be of help to clinicians working with adult ALL patients. However, both tables need to be revised in order to more accurately present the data of the cited papers. Please see the attached file for specific comments on the shortcomings of the manuscript and the tables in particular.

Specific comments

  • Line 36: maybe a more recent citation

Author reply: Thank you. An additional and more recent citation has been added.

  • Line 69: add citation and briefly comment:

o Kyriakidis I, Vasileiou E, Rossig C, Roilides E, Groll AH, Tragiannidis A. Invasive Fungal Diseases in Children with Hematological Malignancies Treated with Therapies That Target Cell Surface Antigens: Monoclonal Antibodies, Immune Checkpoint Inhibitors and CAR T-Cell Therapies. J Fungi (Basel). 2021 Mar 5;7(3):186. doi: 10.3390/jof7030186. PMID: 33807678; PMCID: PMC7999508.

o Lindsay J, Teh BW, Micklethwaite K, Slavin M. Azole antifungals and new targeted therapies for hematological malignancy. Curr Opin Infect Dis. 2019 Dec;32(6):538-545. doi: 10.1097/QCO.0000000000000611. PMID: 31688198.

Author reply: Thank you. Proposed citations were added as requested. These citations were also added in line 189.

  • Line 72: add citation [7]

Author reply: Added as requested.

  • Table 1:

o Heading: use a smaller font size if possible, column title IMI incidence instead of IMIS

Author reply: Thank you. Revised as requested

 o Add a column with the study period (years). Eg the study of Nicolato et el [13] expands over 23 years (1978-2013) vs the one by Koehler et al. that spans over 2 years (2011- 2013)

Author reply: Thank you. Added as requested.

o Antifungal prophylaxis column would be more useful than more specific antimould. Eg in Henden et al [11] fluconazole was used for IFD prophylaxis. Although fluconazole is not considered a mould-active azole, it is useful to include such information in the table, esp when you include it in following studies, such as Doan et al [12]. Also, in the study by Nicolato et al. patients received prophylaxis, incl. mould-active azoles.

Author reply: Thank you. Revised as requested.

o Nicolato et al. study: According to the original paper “Ten episodes (32.2%) occurred in patients receiving prophylaxis with an azole”, not specifically fluconazole

Author reply: Thank you. Revised as requested.

 o Mariette et al. 2017 (not 2016 as in the table): median age 47 not 30 as stated in the abstract. You could include the data on prophylaxis as presented in Table 1 of the original manuscript.

Author reply: Thank you. Revised as requested.

o Keng et al: median age (range) 34 (19-61), these are the demographic characteristics of only the patients with IFI. Similarly, all patients that developed IFI had received prophylaxis (Table 1 of the original manuscript). It’s better to omit “all patients” and the percentages from your table or be more precise in order to avoid confusion. Incidence of Rhizopus spp is the same as for Fusarium (2/209=0.95%).

Author reply: Thank you. Revised as requested.

o Koehler et al: Prophylaxis wasn’t not offered in 42% of the participating centers in the study (8 out of 19). As mentioned in the results section, “a total of 97 patients (54.2%) were receiving antifungal prophylaxis at the time of IA diagnosis.” Please clarify, otherwise it can be interpreted as 42% of the patients did not receive prophylaxis.

Author reply: Thank you. Revised as requested.

 o Cattaneo et al: ALL patients received fluconazole prophylaxis, while AML patients received posaconazole. This info is helpful when reading your comment in the table and later in the discussion. Remove the overall IFI incidence, as you did for the Koehler et study, because they only focus on invasive aspergillosis. Mention that the incidence of IMIs (15.4%) includes proven/probable and possible case. Better include as a comment that “the overall frequency of IPA was similar in AML (22/136, 16.2%) and in ALL patients (6/39, 15.4%).” See my also my comment on line 116.

Author reply: Thank you. Revised as requested.

 o Di Blasi et al: 271 ALL adult patients were included of which 127 had at least one febrile episode. Limited info on antifungal prophylaxis is reported in the manuscript. “Antifungal prophylaxis during remission induction chemotherapy was employed in 37 cases (in 4 cases with posaconazole, in 1 case with itraconazole, in 19 cases with fluconazole, in 13 cases with liposomal amphotericin B).”

Author reply: Thank you. Revised as requested.

 o Valentine et al: 32,815 patients with hematological malignancies were recorder of which 669 cases

Author reply: Thank you. Revised as requested.

o Grundahl et al: According to Table 2 of the manuscript both patients that developed IFD did not receive antifungal prophylaxis and 3 out of 5 patients with probable IFD. Please revise your comment on the table.

Author reply: Thank you. Revised as requested.

  • Lines 109-110: inappropriate reference [26]. Cited paper comments on the EORTC criteria of chemotherapy response in bladder cancer patients.

Author reply: Thank you. Revised as requested.

  • Line 114-116: Cattaneo et al report that “the overall frequency of IPA was similar in AML (22/136, 16.2%) and in ALL patients (6/39, 15.4%)”. And they specify that “relative frequency of proven/probable IPA was higher in ALL (83%) (5 proven/probable and 1 possible) than in AML (54.5%) (12 proven/probable and 10 possible)”. So, most of the IPAs in ALL patients are proven/probable, while almost half of them are possible in the AML patients. Please include all info to draw better conclusions.

Author reply: Thank you. Included as requested

  • Line 119: Specify that 4.7% was the incidence rate of IMIs, because in the previous sentence you comment on the incidence rates of IFIs.

Author reply: Thank you. Revised as requested.

  • Table 2:

o Better title would be “…in hematologic patients”, because most of the cited papers do not focus on ALL patients

Author reply: Thank you. Revised as requested.

o Polymorphisms in genes regulating TNFa production: The cited paper by Sainz et al [29] discusses how downregulation of the TNFa receptor (not TNFa production) increases the susceptibility to IA. They point out that “Although no significant differences were observed in TNFR2 +196 polymorphism between IPA and non-IPA patients (Table 4), VNTR of the TNFR2 promoter was strongly associated with susceptibility to develop IPA.” And later in the paper “It is possible that this variant sequence in the promoter region of the TNFR2 gene decreases expression of TNFR2 and this may explain an increased susceptibility to invasive Aspergillus infection.”

Author reply: Thank you. Revised as requested.

o Polymorphisms in TLR-2 and TLR-4: The reference [30] only highlights the association of TLR-4 (not TLR-2) polymorphisms with IA.

Author reply: Thank you. Revised as requested.

o COPD: Reference [34] studies the incidence of IPA in COPD patients. No mention of COPD as a comorbidity of hematologic patients in this study by Guinea et al. In their analysis they included patients that were admitted to the hospital with a COPD diagnosis ± IPA. A more appropriate citation is needed.

Author reply: We added reference 36 to further support our claim.

o Smoking: The authors in the cited letter [35] found that tobacco and marijuana are heavily contaminated with fungal spores. However, they didn’t show any data comparing the incidence of IMIs in smokers versus non-smokers. A more appropriate citation is needed.

Author reply: Thank you. We added reference 36 to further support our claim.

o Lower probability of CR: According to ref. 36 WBC more or equal (not >) 30000/μL (not ML).

Author reply: Thank you. Revised as requested.

o Nucci and Anaissie in their opinion paper (ref. [36]) cite [34] and [35] and list COPD and smoking as risk factors of IFDs. Similarly, they cite other studies to support that according to them the lower probability of CR, etc are additional risk factors. You could also add this reference above to support you claim that smoking and COPD are risk factors of IFDs and IMIs.

Author reply: Thank you. Revised as requested.

  • Line 151: specify that the D-index was developed to measure the risk of IMIs in AML patients.

Author reply: Thank you. Revised as requested.

  • Lines 175-176: As per ref. [48] “breakthrough fungal infection was reported in 16/21 (75%), 14 (65%) patients had CAPA while on echinocandin prophylaxis, while 2 (10%) patients had CAM while on voriconazole prophylaxis”, so 65% was on echinocandin (micafungin or anidulafungin according to Table 3) prophylaxis upon CAPA diagnosis.

Author reply: Thank you. Revised as requested.

  • Line 266: Move ref [63] at the end of the previous sentence on line 263

Author reply: Thank you. Revised as requested.

  • Line 323: Correct the percentage and put the equal to sign after p (7.9% vs 11.7% p=0.24) “Rates of proven and probable IFD assessed independently were 7.9% (18/228) in the L-AMB group and 11.7% (13/111) in the placebo group (p = 0.24).

Author reply: Thank you. Revised as requested.

Reviewer 3 Report

The manuscript is a Review about acute lymphoblastic leukemia (ALL) and invasive mould infections. The review addressed the main factors about ALL, treatment of fungal infections in patients with ALL, most used drugs, drug interactions, new drugs, among others. This paper adds to the literature information on various topics related to fungal infections associated with ALL.

Suggestions for improved presentation and visualization of results were given. The paper must be returned to the authors in order to clarify any doubts and  to make some suggested changes.

Minor comments:

1. Line 1, “title” –  in several parts of the article, data on fungal infections are referenced and not just mold infections. Thus, at the authors' discretion, the title could include the term “invasive fungal infections” instead of “invasive mold infections”.

2. Table 1, Lines 354, “spp” and “spp.” – please heed the journal's standards and standardize the term throughout the manuscript.

3. Table 1 – This table is important however needs an improvement in its layout.

Author Response

Dear Reviewer,

Thank you for reviewing our manuscript and for giving us an opportunity to revise our work. We have revised our manuscript jof-1979500 entitled “Acute lymphoblastic leukemia and invasive mould infections. A challenging field” in accordance with your comments. You can view all changes in the revised manuscript using the “Track Changes” function.

We would like to thank you for your time spent reviewing our manuscript and for your helpful and constructive comments which have improved it. We have carefully addressed, point-by-point, the concerns raised in the comments by you. Our responses are highlighted in red as follows.

The manuscript is a Review about acute lymphoblastic leukemia (ALL) and invasive mould infections. The review addressed the main factors about ALL, treatment of fungal infections in patients with ALL, most used drugs, drug interactions, new drugs, among others. This paper adds to the literature information on various topics related to fungal infections associated with ALL.

Suggestions for improved presentation and visualization of results were given. The paper must be returned to the authors in order to clarify any doubts and to make some suggested changes.

  • Line 1, “title” – in several parts of the article, data on fungal infections are referenced and not just mold infections. Thus, at the authors' discretion, the title could include the term “invasive fungal infections” instead of “invasive mold infections”.

Author reply: Thank you for your comments. We agree with you that there is lack of data regarding IMIs in the setting of ALL and that is why we comment on this throughout the manuscript. The scope of this review is to shed light on current data regarding IMIs in ALL patients, so we thought that the title we chose would be more representative of our purpose.

  • Table 1, Lines 354, “spp” and “spp.” – please heed the journal's standards and standardize the term throughout the manuscript.

Author reply: Thank you. Revised as requested.

  • Table 1 – This table is important however needs an improvement in its layout.

Author reply: We fully agree with your comment. Table 1 has been modified according to yours and other reviewers’ comments.